# Effects of Forest on Birdsong and Human Acoustic Perception in Urban Parks: A Case Study in Nigeria

**Mary Nwankwo [1], Qi Meng [1,2,\*], Da Yang [1] and Fangfang Liu [1]**

[1] Key Laboratory of Cold Region Urban and Rural Human Settlement Environment Science and Technology, Ministry of Industry and Information Technology, School of Architecture, Harbin Institute of Technology, 66 West Dazhi Street, Nan Gang District, Harbin 150001, China; blessedmaryc@gmail.com (M.N.); da.yang@hit.edu.cn (D.Y.); liufangfang@hit.edu.cn (F.L.)

[2] Key Laboratory of Architectural Acoustic Environment of Anhui Higher Education Institutes, Anhui Jianzhu University, Hefei 230601, China

[\*] Correspondence: mengq@hit.edu.cn

**Abstract:** The quality of the natural sound environment is important for the well-being of humans and for urban sustainability. Therefore, it is important to study how the soundscape of the natural environment affects humans with respect to the different densities of vegetation, and how this affects the frequency of singing events and the sound pressure levels of common birds that generate natural sounds in a commonly visited urban park in Abuja, Nigeria. This study involves the recording of birdsongs, the measurement of sound pressure levels, and a questionnaire evaluation of sound perception and the degree of acoustic comfort in the park. Acoustic comfort, which affects humans, describes the fundamental feelings of users towards the acoustic environment. The results show that first, there is a significant difference between the frequency of singing events of birds for each category of vegetation density (low, medium, and high density) under cloudy and sunny weather conditions, but there is no significant difference during rainy weather. Secondly, the measured sound pressure levels of the birdsongs are affected by vegetation density. This study shows a significant difference between the sound pressure levels of birdsongs and the vegetation density under cloudy, sunny, and rainy weather conditions. In addition, the frequency of singing events of birds is affected by the sound pressure levels of birdsongs with respect to different vegetation densities under different weather conditions. Thirdly, the results from the respondents (N = 160) in this study indicated that the acoustic perception of the park was described as being pleasant, vibrant, eventful, calming, and not considered to be chaotic or annoying in any sense. It also shows that the human perception of birdsong in the park was moderately to strongly correlated with different densities of vegetation, and that demographics play an important role in how natural sounds are perceived in the environment under different weather conditions.

**Keywords:** soundscape; urban park; birdsong; natural vegetation

## 1. Introduction

Various studies in Nigeria have explored the benefits of green areas and how urban areas can be improved. Likewise, different approaches have been adopted to further tackle the challenges that are faced in improving urban areas. Results have shown that the improvement of urban areas in Nigeria is negatively affected by the rapid rate of urbanization, less attention being paid to continuous development, and a limited budget for the physical planning and maintenance of green areas [1]. The importance of green areas in cities has been identified in various ways since the 19th century. Green areas reduce urban air pollution and provide environmental, social, and economic value to society [2]. Because of its importance, the green city concept was initiated in Nigeria to focus on achieving sustainable, green, and environmentally friendly cities, and to reduce the negative effects caused by deforestation, especially the emission of $CO_2$ into the atmosphere [3]. Another

important aspect of a functioning city is its urban forest [4,5] and green spaces [6]. This is because green spaces provide an extensive range of benefits, such as climate adaptation [7], climate mitigation [8,9], erosion control, and physical and psychological comfort [10], and because they affect human senses such as sight, smell, sound, taste, and touch [11]. Research by [12] using a quantitative method studied the variance between urban greenery, urban development, and the quest for environmental sustainability in northern Nigerian cities, and the scientific findings indicated that the allocation of open green spaces has not been harmonized with the urban population. This is influenced by a low percentage of urban greenery. This research further suggested the need for a strict adherence to sustainable urban planning that would integrate physical development and environmental considerations in order to enhance urban greenery. Further strategies that support architects and urban planners in developing guidelines required for city planning and design were suggested by [13,14]. These strategies are of high value to the inhabitants, as they support social meetings for all ages and promote the continuous growth and development of urban areas. Although the importance of green spaces and the need for the proper articulation and implementation of planning policies has been constantly emphasized in Nigeria, more studies need to be conducted on individual green spaces, urban parks, and forest and recreational centers that contribute to the physical, social, and health development of urban areas.

Urban parks are recognized as being major contributors to the physical and aesthetic qualities of urban centers, and they play a role in improving the quality of living in urban areas [15–17]. Their uses are good, and generally, humans are free to engage in healthy exercise or other recreational activities. This in turn promotes health, restorative life [18], and better social interactions [19,20]. Urban parks, relaxation gardens, and healing gardens are perfect scenes where natural soundscapes can be perceived [18], and they can offer a reduced degree of exposure to the adverse effects of anthropogenic noise in urban areas [21]. The benefits of urban parks possess high societal values that serve as pathways for economic growth and locations for the complex network of recreational activities that are essential to human function and standards of living. In addition, they are dynamic places with changeable environments that can transform man's idea of nature, and they fulfill a variety of human needs, such as better air quality and noise reduction [22]. They also serve as a useful environmental source for urban dwellers to improve their physical, mental, cognitive, and social wellbeing [23,24]. Parks can consist of natural vegetation, which is a source of natural sounds, and they can function as an important part of the natural ecosystem. When natural scenes are dominated by green vegetation, it has been studied as a restorative environment [25] where exposure provides restoration from stress and mental fatigue [26]. As the human population increases, especially in urban areas, urban dwellers are constantly faced with stress and the need to immerse themselves daily in nature, and to perceive natural sounds. It is the right moment to address factors that affect common avian species in Nigeria and to consider the utmost need for urban bird conservation. Perceived natural sounds are important and useful signals that improve our daily activities. They are part of the natural landscape and play a huge role in the ecosystem [27]. Natural sounds are vital cues that are used by humans to communicate with one another and to perceive environmental conditions [28]. The study of soundscapes has evolved through various disciplines, such as anthropology, acoustics, architecture, ecology, psychology, and landscaping [29,30]. However, it was originally rooted in music, and was first defined by Murray Schafer as any acoustic field of study [31]. Recently, the International Standard Organization (ISO) defined a soundscape as the acoustic environment as perceived, experienced, or understood by a person or people in context [32]. As the study of sound expanded to other disciplines, research into its relationship with the landscape [33] emerged, where it is used to denote the overall sonic environment. The soundscape and the acoustic comfort of an open area or a closed space, such as a classroom, can be designed, measured, and evaluated [34,35]. This can be achieved by using various methods, including laboratory experiments and questionnaires [5], sound walks [36], simulations and virtual

modeling [37,38], and psychoacoustic parameters [36,39]. Soundscape research has shown that natural sounds are commonly perceived as being pleasant by humans [40,41] and they have a positive effect on quick recovery from psychological and physiological stress [19]. Researchers such as [5] have contributed to the growing knowledge of soundscapes by evaluating soundscape perception and preferences amongst different users in an urban recreational forest park in Xi'an. The studies revealed that natural sounds were perceived more positively than other artificial sounds that appeared to be dominating in the park, with age and gender also playing important roles as to how certain sounds were tolerated or perceived with different levels of sensitivity.

According to some previous studies, birds are referred to as being one of the most important types of animals that generate pleasant sounds [42,43]. These sounds allow for individual experiences in nature, and they are exceptionally rich in semantic values and associations [44]. Bird sounds possess symbolic values, which affect how they are cognitively appraised and how restorative they are perceived to be [45,46]. Most of the research on the restoration of natural soundscapes in parks and green areas has been focused on birdsong [47]. For instance, research by [48] explained that the soundscape of a park with rich bird sounds can minimize the adverse effects of traffic noise, provide nature-based solutions to human health, and improve general wellbeing in urban areas. In addition, a field experiment by [49] with 70 participants in Shenyang, China found that natural environments with natural sounds have positive effects on the restoration of individual attention. In situations where a space's soundscape is created by birds, it will have peaks of intensity, quantity, and diverse frequencies. Additionally, the time of the year, the hour of the day, meteorological conditions, climate dynamics [27,29], landscape structure, the structure of vegetation [50,51], and human disturbances are factors that could affect bird activities [52]. It has been found that birdsong increases positive perceptions in humans, as well as reducing psychological stress, and it can also be affected by natural factors [19,45,53]. Thus, this paper also provides its own perspective on the factors that affect birds and the generation of birdsongs in urban parks.

Despite research on the promotion of sustainable urban environments and the evaluation of users' perceptions of green spaces in Nigeria, few studies have been conducted specifically regarding natural soundscapes in green spaces, how the natural vegetation can affect natural sounds, human acoustic comfort, and the effects of change in weather conditions on humans and birds in commonly visited urban parks in Nigeria. Thus, this paper aims at exploring natural soundscapes through the study of birdsongs in different vegetation densities and how it affects humans. It focuses on the following research questions: (1) whether there is a significant difference between the density of natural vegetation, the frequency of singing events, and the singing duration of the birds in an urban park under different weather conditions; (2) whether the density of natural vegetation has a significant difference in the sound pressure level of the birdsongs under different weather conditions in an urban park; and (3) whether human acoustic perception in the park is appropriate, especially when listening to the birdsongs. In addition, what are humans' preferences for weather conditions when visiting the park, and the preferences for natural vegetation density? Therefore, this study characterized different vegetation densities in an urban park into three groups: low density, medium density, and high density, before selection. In addition to habitat, preferences such as the presence of evergreen trees for nesting and the presence of food for the birds was taken into consideration over a period of time. The study also analyzed the frequency of singing events of five common bird species in the park, and furthermore, involved in situ sound recordings, measurements, and a questionnaire survey.

## 2. Methodology

### 2.1. Study Area

This study area is located within Abuja city in Nigeria (Figure 1). The park was selected due to current urban development activities around it, with Abuja being one of the fastest-

growing cities on the African continent. Abuja city lies within a latitude of 8°25″–9°25″ N and longitude of 6°45″–7°45″ E, and is 180 feet above sea level and 8000 km² in land mass [54]. The territory is bounded by Niger to the west and northwest, Kaduna to the northeast, Nasarawa to the east, and Kogi to the southwest, with an estimated population of 6 million as of 2016 [55].

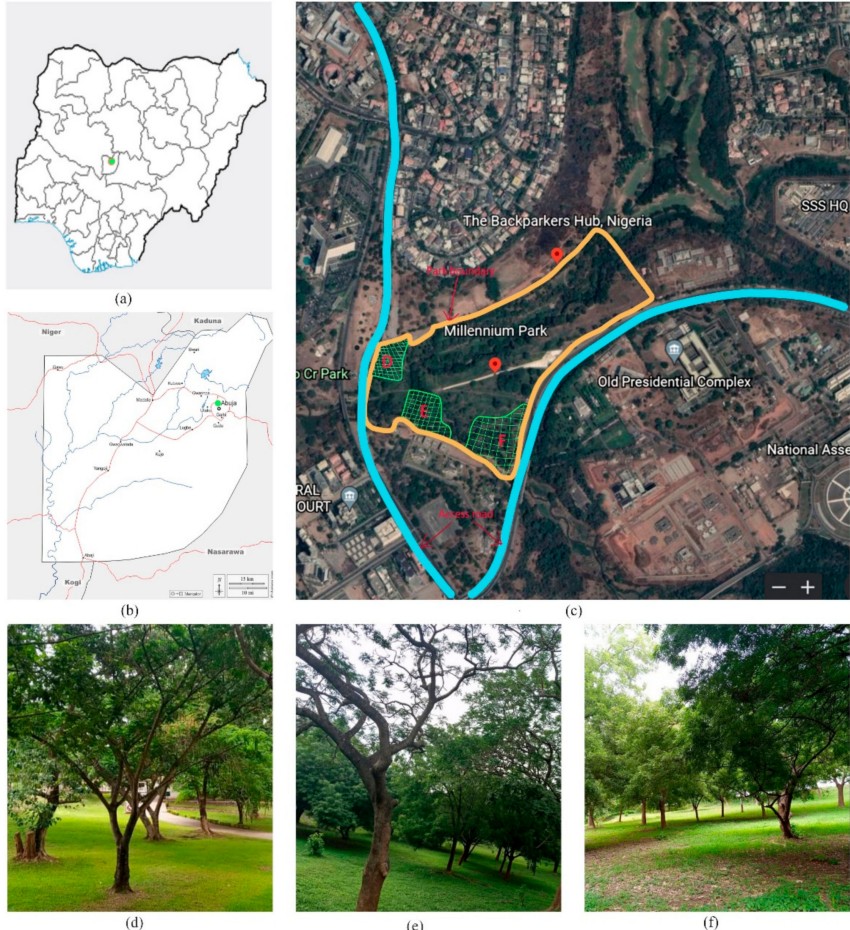

**Figure 1.** Map and aerial photographs of the study area. (**a**) Map of Nigeria showing Abuja; (**b**) map of Abuja showing Maitama; (**c**) Google Earth map of the study area; (**d**) low-density vegetation; (**e**) medium-density vegetation; (**f**) high-density vegetation.

The study park is Millennium Park in the Federal Capital Territory, Abuja. It is known as the largest public park located in the Maitama district in the capital city of Nigeria. The park was established in 2003, has a total area of 80 acres (32 ha), and is bounded by two major traffic roads in the city. It is the largest multifunctional place of entertainment, allowing for leisure, physical fitness, and other forms of social activities. It attracts a large number of visitors due to its size and natural landscapes; the park is separated by a river along its main rectilinear axis. Its untouched natural vegetation, comprising large evergreen trees that provide shelter and that create a serene environment, distinguishes it from other parks, and it houses different kinds of tropical birds and other natural creatures. Considering that the potential effects of urban noise may influence the birds' sounds [56], the case study site was therefore selected to be distant from traffic roads and residential areas to avoid urban noise. This site is characterized by mountainous vegetation, deciduous forest, savanna, and brushwood vegetation types, with species of flower-bearing plants such as Japanese jasmine (*Jasminum mesnyi*), Arabian jasmine (*Jasminum sambac*), trumpet jasmine (*Jasminum bignoniaceum*), dwarf poinciana (*Caesalpinia pulcherrima*), maidenhair fern (*Adiantum* spp.), and coleus (*Solenostemon*). It also consists of capa de obispo (*Acalypha*

*wilkesiana*), zedoary (*Curcuma zedoaria*), canna (*Canna indica*), broadleaf palm-lily (*Cordyline fruticosa* L. *A. Chev*), and pigeon berry (*Duranta repens*). Trees include the southern catalpa (*Catalpa bignonioides*), pacara earpod tree (*Enterolobium contortisiliquum*), Malayan banyan (*Ficus microcarpa*), masquerade tree (*Polyalthia longifolia*), sea randa (*Guettarda speciosa*), neem tree (*Azadirachta indica*), flamboyant tree (*Delonix regia*), Broome raintree (*Albizia lebbeck*), Lombardy poplar (*Populus nigra*), weeping willow (*Salix babylonica*), common spruce (*Picea*), Siberian elm (*Ulmus pumila*), and grasses include Bermuda grass (*Cynodon dactylon*).

### 2.2. Measurement of Sound Pressure Levels

Five different species of birds were identified in each density level of vegetation (low density, medium density, and high density), namely: 1. African reed warblers (*Acrocephalus baeticatus*); 2. African melodious warblers (*Hippolais polyglotta*); 3. wild doves (*Zenaida macroura*); 4. sparrows (*Spizella passerina*); and 5. black kites (*Milvus migrans*). In situ measurements were conducted in the three vegetation densities with respect to the time of the day and the three different weather conditions (cloudy, sunny, and rainy). The different temperatures of the day were recorded for each day, with cloudy days at 26 °C, sunny days at 32 °C, and rainy days at 24 °C. The sound pressure level was measured at specific times of the day between the morning hours (8:00 a.m.) and the evening hours (18:00 p.m.) at 15 min intervals. The sound pressure level was measured with a digital sound level meter (AS824; China), an instrument comprising a microphone that captures and accesses sound by measuring sound pressure levels in decibels (dBA). Slow, fast, and impulse are the three types of sampling settings, depending on the intended results. Since bird sounds usually change quickly and can occur within a short period of time, the digital sound level meter was set to fast sampling [57]. The instruments were placed at different locations in each habitat below the singing birds, at a vertical distance of 3 m, and measurements were recorded with respect to the different times of the day.

### 2.3. Sound Recording

The birdsongs were recorded to determine the frequencies of singing events for the five different species of birds identified. Each recording was conducted using a H2n Handy Recorder (Zoom Corporation) in the low-density, medium-density, and high-density vegetation areas of the park. The recordings were made under cloudy, sunny, and rainy weather conditions in the park, between the hours of 8:00 and 18:00, at one-hour intervals. In order to record better quality bird sounds from different directions, an omnidirectional microphone was attached to the recording instrument (H2n Handy Recorder) [58], which was placed vertically below the singing birds at 3 m above ground level with the aid of a tripod. Raptor birds such as black kites could be seen hovering in the sky and perching at intervals in the trees. However, the birds could also perch in the trees for calls or for rest, and this was when their songs were comfortably recorded at a height of 3 m. All of the recordings were conducted simultaneously with H2n recorders in the low-density, medium-density, and high-density vegetation areas. Previous studies have shown that crowd density may have an influence on the vocalizations of birds [59]. Measurements of crowd density in the low-, medium-, and high-density vegetation areas were conducted, and the change in crowd density had no significance difference in the vocalizations of the birds among the three categories of vegetation. Therefore, a change of crowd density may not affect the comfortable vocalizations of the birds.

### 2.4. Questionnaire Design

A questionnaire survey is a tool that can be used to describe the perceived or experienced acoustic environment [60]. A web-based and physical questionnaire containing 15 questions in total was designed for data collection. A total of 171 questionnaires were sent out, adopting an evaluation scale from 1 to 5 (a 5-point Likert scale). A total of 160 effective questionnaires were retrieved, with an effective rate of 93.57%, and a total of 83 males and 77 females participated. The questionnaire questions were divided into four sections:

(1) demographics and social information; (2) the perception of acoustic comfort when listening to birdsongs in the park; (3) the preference for different weather conditions to visit; and (4) the preference for vegetation density areas to remain in. The first section was designed to capture demographic data, including the age, gender, and educational qualifications of the respondents. The second section was designed to obtain data on the perception of acoustic comfort using six indicators [61]: pleasant, calming, annoying, eventful, chaotic, and vibrant. The third section was designed to obtain a database of the preferred weather conditions for visiting in the park, and the fourth question was designed to obtain data based on the preferred density of vegetation in the park. The ages of the respondents were divided into six groups [62,63], and Table 1 shows the questionnaire questions. It took approximately 5–10 min for each respondent to complete the questionnaire.

**Table 1.** Demographic and social data, and acoustic perceptions determined by the questionnaire.

| Demographic and Social Indicators | Categorization and Scale |
| --- | --- |
| Gender | 1: male; 2: female |
| Age | 1: <18; 2: 19–30; 3: 31–40; 4: 41–50; 5: 51–60; 6: >60 |
| Educational level | 1: primary; 2: secondary; 3: indergraduate; 4: graduate; 5: postgraduate |
| Questions on the acoustic perception of the sound environment and birdsongs in the park | Strongly disagree, disagree, neither disagree nor agree, agree, strongly agree |
| To what extent do you perceive the current sound environment and birdsongs as being pleasant? To what extent do you perceive the current sound environment and birdsongs as being calming? To what extent do you perceive the current sound environment and birdsongs as being annoying? To what extent do you perceive the current sound environment and birdsongs as being eventful? To what extent do you perceive the current sound environment and birdsongs as being chaotic? To what extent do you perceive the current sound environment and birdsongs as being vibrant? | |
| Questions on weather conditions when visiting the park | <1 h, 1 h, 2 h, 3 h, >3 h |
| How long can you visit the park on a cloudy day? How long can you visit the park on a sunny day? How long can you visit the park on a rainy day? | |
| Questions on the density of vegetation when relaxing in the park | 1 h, 1 h, 2 h, 3 h, >3 h |
| How long can you be in the park in a low-density vegetation area? How long can you be in the park in a medium-density vegetation area? How long can you be in the park in a high-density vegetation area? | |

*2.5. Data Analysis*

The software used for the statistical analysis of the collected data was SPSS 26.0 (IBM, Armonk, NY, USA) and Microsoft Excel. This study adopted common statistical methods. A one-way analysis of variance (ANOVA) was used to ascertain whether there were any significantly significant differences between the frequency of singing events and the sound pressure levels of the birdsongs in low, medium, and high densities of vegetation. The null hypothesis $H_0$ states that there is no significant difference between each group, and the alternative hypothesis $H_A$ states that there is a significant difference between each group at a 95% confidence level. In addition, Tukey's honest significance test statistically determined whether the low-density, medium-density, and high-density vegetation groups were significantly different to each other, and which were not. Linear regression determined the association between the sound pressure level and the frequency of singing events, the chi-squared test determined the relationships and significance between the age of the respondents and the preference for weather conditions, and Spearman's rank correlation showed the strength and correlation between each of the perceived acoustic indicators and each density level of the vegetation.

## 3. Results

### 3.1. Effects of Different Densities of Vegetation on the Singing Events of Birds

A one-way analysis of variance (ANOVA) based on the stated null and alternative hypothesis showed whether there were any statistical differences between the frequency of singing events of the birds in low-density, medium-density, and high-density vegetation under the three different weather conditions at a 95% confidence level. The analysis of variance results were as follows: the frequency of birdsong in low-density, medium-density, and high-density vegetation on a cloudy day was $F_{(2, 27)} = 3.47$, $p = 0.045$; on a sunny day, $F_{(2, 27)} = 3.40$, $p = 0.048$; and on rainy day, $F_{(2, 27)} = 2.43$, $p = 0.146$. The results indicated that there is a significant difference between low-, medium-, and high-density vegetation and the frequency of singing events (frequency). Although this difference was only obtainable on a cloudy day and on a sunny day, with $p$-values of $p = 0.045$ and $p = 0.048$, respectively, it showed a non-significant effect on a rainy day, with a $p$-value of $p = 0.146$.

Since the one-way ANOVA is an omnibus test, Tukey's honest significance test (Table 2) was performed on variables that showed significant differences to identify the actual vegetation density that was statistically significant to each other and that which was not. The results indicate that there was a significant difference between the frequency of the birdsongs in low-density vegetation and high-density vegetation on a cloudy day, with $p = 0.036$, but no significant difference between the medium- and high-density vegetation areas, with $p = 0.036$ on the same day. In addition, there was a significant difference between low-density and high-density vegetation on a sunny day, with $p = 0.042$, but there was no significant difference between the medium-density and high-density categories.

**Table 2.** Measures of frequency of birdsong events in the three densities of vegetation in the park, where LDV = low–density vegetation, MDV = medium–density vegetation, and HDV = high–density vegetation.

| Density of Vegetation | | Mean Difference | *p*-Value | Upper Bound | Lower Bound |
|---|---|---|---|---|---|
| | | **Cloudy Day** | | | |
| MDV | LDV | 19.20 | 0.339 | −14.04 | 52.44 |
| | HDV | −16.10 | 0.463 | −49.34 | 17.14 |
| HDV | LDV | 35.30 * | 0.036 * | 2.06 | 68.54 |
| | | **Sunny Day** | | | |
| MDV | LDV | 10.50 | 0.668 | −19.67 | 40.67 |
| | HDV | −20.70 | 0.223 | −50.87 | 9.47 |
| HDV | LDV | 31.20 * | 0.042 * | 1.03 | 61.37 |

* Correlation is significant at $p < 0.05$.

The five species of birds identified in the low-, medium-, and high-density vegetation areas were: African reed warbler (*Acrocephalus baeticatus*), African melodious warbler (*Hippolais polyglottal*), black kite (*Milvus migrans*), wild dove (*Zenaida macroura*), and sparrow (*Spizella passerina*). The songs of the above listed birds were heard in the low-, medium-, and high-density vegetation areas on cloudy, sunny, and rainy days at different times of the day, from 8:00 to 17:00 at 1 h intervals, as shown in Figure 2a–e. The observed non-polygynous African reed warbler birds sang for a longer period (8 h) on a cloudy day in low-density vegetation, and they had no singing time in medium-density vegetation on the same day. However, it was observed that the African reed warblers had frequent singing times within all densities of vegetation on sunny and rainy days, as shown in Figure 2a. Likewise, the African melodious warbler birds experienced their longest singing time of a duration of 5 h in low-density vegetation on a rainy day, but they experienced no singing time in medium-density vegetation on a sunny day or in high-density vegetation on a rainy day, as shown in Figure 2b. Black kites are normally medium-sized prey birds that are generally dark in color and they are observed around areas with water bodies, where they have access to their prey. This also explains their presence in the park, as the park is characterized by

the presence of a river. Since they are raptor birds, they usually hover in the sky, perching at intervals and monitoring the ground prey. The longest singing period of a black kite was recorded on a rainy day, for a duration of 9 h, in medium-density vegetation. Additionally, it was observed that most of the calls of the black kites were recorded toward afternoon and evening, with few calls being recorded in the morning, as shown in Figure 2c, and few calls being recorded on a sunny day in medium-density vegetation. No calls were recorded on a rainy day in low-density vegetation. The black kite call duration was between 8 and 10 times within 30 s.

The wild dove calls (Figure 2d) were recorded in all three densities of vegetation, on cloudy, sunny, and rainy days, with the highest singing duration of 9 h recorded in medium-density on a sunny day, and the lowest singing duration of 2 h recorded in medium-density on a rainy day. One of the characteristics of wild doves in Nigeria is that they are commonly observed birds, especially in evergreen natural vegetation and in areas that are characterized with flamboyant and neem trees, which were present in the study area. Wild doves are easily spotted in urban or rural areas of Nigeria. The wild dove call duration was between seven and eight times within 20 s. The chipping sparrows are found in shrubs or in trees in houses where they are free to nest. It was observed that the same singing period of 5 h was recorded for sparrows on sunny and rainy days in high-density vegetation, and the lowest singing duration of 1 h was recorded in low-density vegetation on a sunny day, as shown in Figure 2e. There were no calls in low-density vegetation on cloudy and rainy days, respectively. From the observations, it can be seen that each species of bird experienced a different duration of calls in different densities of vegetation. From this study, it can be seen that the wild doves had the highest singing period, followed by black kites, African reed warblers, sparrows, and African melodious warblers. In addition, more birdcalls were recorded in higher density vegetation, followed by medium-density vegetation and low-density vegetation.

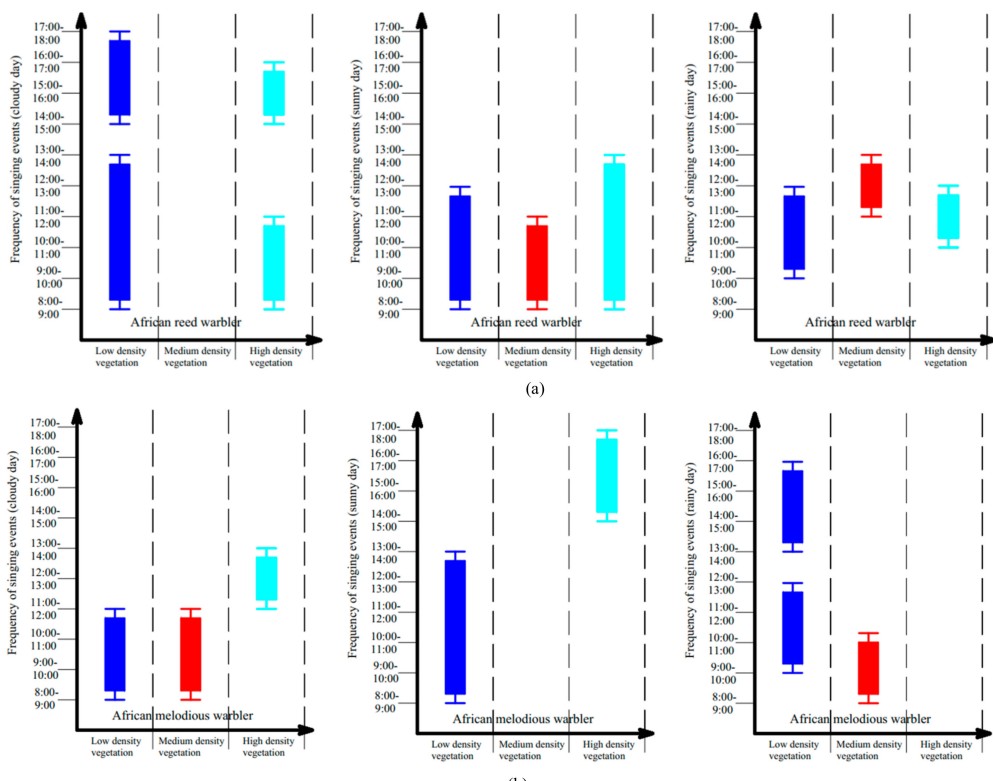

**Figure 2.** *Cont.*

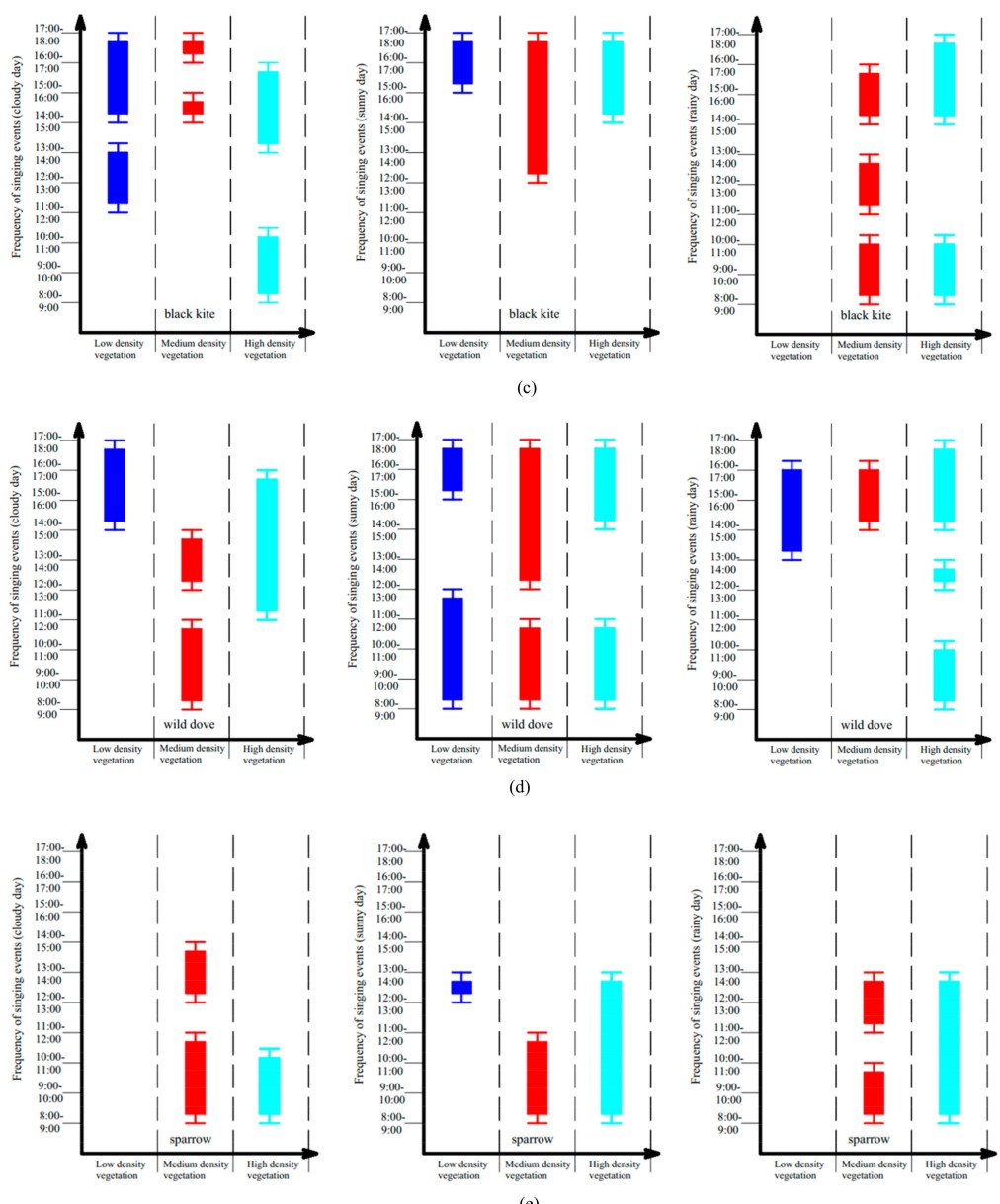

**Figure 2.** Specific singing duration of the birds in low–, medium-, and high–density vegetation. (**a**) African reed warbler; (**b**) African melodious warbler; (**c**) black kite; (**d**) wild dove; and (**e**) sparrows.

*3.2. Effect of Frequency of Singing Events on Sound Pressure Levels of Birdsongs*

The measured sound pressure levels of the birdsongs were measured in each density of vegetation on a cloudy day, sunny day, and rainy day, and the data were analyzed using a one-way analysis of variance. The findings showed a significant difference between the sound pressure levels of the birdsongs in each density of vegetation. The statistical results were as follows: for low-density vegetation, $F_{(2, 27)} = 11.15$, $p = 0.000$ ($p < 0.01$), for medium-density vegetation, ($F_{(2, 27)} = 4.49$, $p = 0.021$, and for high-density vegetation, $F_{(2, 27)} = 3.44$, $p = 0.047$ ($p < 0.05$). It can be explained that the density of vegetation in which the birds are located can affect their sound pressure levels because they are responsive to their environment. It can also be explained that they also react based on the weather conditions of the day. For a further analysis and for an in-depth study, the relationship between the frequency of the singing events of the birds and the sound pressure level of the birdsongs was analyzed. Figure 3a–i shows the graphical representation of the results obtained. Figure 3a shows that on a cloudy day in low-density vegetation,

there was an excellent relationship between the frequency of singing events and the sound pressure levels of the birds, with a correlation coefficient of $R = 0.75$ and a coefficient of determination of $R^2 = 0.57$. In total, 57% was affected by the sound pressure level of the birdsong, whereas 43% could be attributed to other factors that could affect the birds. However, on the same cloudy day in high-density vegetation, the correlation coefficient ($R = 0.64$) implies a moderate relationship, with 41% attributed to sound pressure levels and 59% attributed to factors other than the frequency of singing events, as shown in Figure 3c. The highest correlation coefficient ($R^2 = 0.82$) was obtained in medium-density vegetation on a sunny day, indicating an excellent relationship between the frequency of singing events and the sound pressure levels of the birdsongs shown in Figure 3e, with only 18% being attributed to factors other than the frequency of singing events. Figure 3g shows that in low-density vegetation on a rainy day, the correlation coefficient ($R = 0.89$) obtained also implies that there is an excellent relationship between the frequency of singing events of birds and the sound pressure level of the birdsongs. A correlation coefficient of $R^2 = 0.79$ showed that 79% was attributed to the sound pressure level and 21% was attributed to other factors. In addition, Figure 3h also shows that a coefficient of determination of $R^2 = 0.47$ was obtained, indicating that 47% of the sound pressure level affects the singing times of birds, whereas 53% is attributed to other factors. However, Figure 3i showed a weak relationship between the sound pressure level and the frequency of singing events in the birds. This indicated ($R^2 = 0.10$) that 10% was affected by the sound pressure level, whereas 90% was attributed to factors other than the frequency of singing events in high-density vegetation on rainy days.

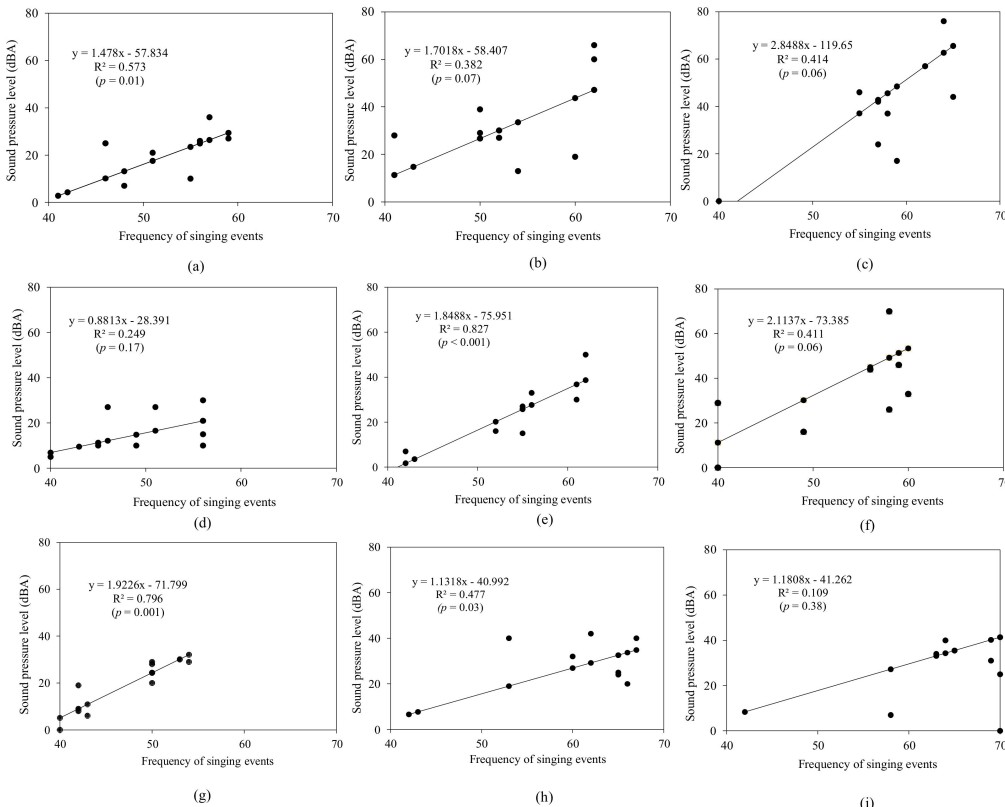

**Figure 3.** Relationship between the frequency of singing events and the sound pressure levels of birdsongs. (**a**) Low–density vegetation on cloudy days; (**b**) medium–density vegetation on cloudy days; (**c**) high–density vegetation on cloudy days; (**d**) low–density vegetation on sunny days; (**e**) medium–density vegetation on sunny days; (**f**) high–density vegetation on sunny days; (**g**) low–density vegetation on rainy days; (**h**) medium–density vegetation on rainy days; and (**i**) high–density vegetation on rainy days.

### 3.3. Effects of Densities of Vegetation and Weather Conditions on Acoustic Perception

The arithmetic mean values of the perceived acoustic comfort when listening to birdsongs in a park, the preference for visiting under different weather conditions, and the preference of density of vegetation to stay in were analyzed and are shown in Table 3. In addition, the results from the demographic information show that 51.9% of the respondents were male and 48.1% were female, and the ages of the respondents were <18 (6.9%), 19–30 (53.8%), 31–40 (20.6%), 41–50 (10%), 51–60 (7.5%), and >60 (1.3%), as shown in Table 4. The results of the mean and standard deviation of the ages of the respondents were(mean = 2.61; SD = 1.087, where N = 160.

**Table 3.** Arithmetic mean values of human acoustic perception and preferences for weather conditions and vegetation densities.

| Perceived Acoustic Comfort Indicators | Arithmetic Mean Values | | |
| --- | --- | --- | --- |
| | Mean | SD | SE Mean |
| Pleasant | 3.55 | 1.17 | 0.09 |
| Calming | 3.55 | 1.09 | 0.86 |
| Annoying | 2.07 | 0.79 | 0.63 |
| Eventful | 4.03 | 0.74 | 0.05 |
| Chaotic | 2.09 | 0.66 | 0.83 |
| Vibrant | 3.82 | 0.74 | 0.58 |
| **Preference of Different Weather Conditions** | **Arithmetic Mean Values** | | |
| | Mean | SD | SE Mean |
| Cloudy day | 2.37 | 1.23 | 0.09 |
| Sunny day | 2.80 | 1.28 | 0.10 |
| Rainy | 1.45 | 1.20 | 0.09 |
| **Preference of Different Densities of Vegetation** | **Arithmetic Mean Values** | | |
| | Mean | SD | SE Mean |
| Low density | 3.03 | 1.31 | 0.10 |
| Medium density | 2.90 | 1.20 | 0.09 |
| High density | 2.68 | 1.73 | 0.13 |

**Table 4.** Demographic information and social factors of the respondents.

| Demographics | | Percentage (%) |
| --- | --- | --- |
| **Gender** | Male | 51.9 |
| | Female | 48.1 |
| Age | <18 | 6.9 |
| | 19–30 | 53.8 |
| | 31–40 | 20.6 |
| | 41–50 | 10 |
| | 51–60 | 7.5 |
| | >60 | 1.3 |
| Educational background | Secondary school | 2.5 |
| | Undergraduate | 28.8 |
| | Graduate | 50 |
| | Postgraduate | 18.7 |

The results of the mean and standard deviation for perceived acoustic comfort were pleasant, M = 3.55; SD = 1.17, calming M = 3.55; SD = 1.09), annoying M = 2.07; SD = 0.79, eventful M = 4.03; SD = 0.74, chaotic M = 2.09; SD = 0.66, and vibrant M = 3.82; SD = 0.74. Based on the scale of the analysis of the result, the mean values for pleasant, calming, eventful, and vibrant indicated that the respondents positively agreed with the degree of acoustic comfort in the park when listening to the birdsongs. In addition, the mean

values of annoying and chaotic indicated that the respondents disagreed that the birdsongs in the park were annoying sources of noise, or were perceived as unpleasant. However, the mean values for the degree of human comfort in different weather conditions, which were cloudy, M = 2.37, sunny, M = 2.80, and rainy, M = 1.45, indicate that the respondents preferred to spend more time in the park on sunny days, followed by cloudy days, and the least time on rainy days. Likewise, the mean values of the degree of comfort in different densities of vegetation were: low density, M = 3.03, medium density, M = 2.90, and high density, M = 2.68. This indicated that the respondents preferred to stay or spend more time in low-density vegetation, followed by medium-density vegetation, and the least time in high-density vegetation within the park. Figure 4 shows the duration of stay for the respondents under different weather conditions (cloudy, sunny, and rainy) in the park. For periods of less than one hour (<1 h), 83.3% of the respondents indicated that they spent the least time in the park on a rainy day. Additionally, 45% of the respondents spent less than one hour on a cloudy day, and 12.5% on a sunny day. For longer durations of stay (>3 h), the least percentage of respondents recorded was on rainy days with 2.5%, followed by cloudy days (8.8%) and sunny days (10%). Generally, more respondents spent longer hours on a sunny day, followed by cloudy days, and the shortest duration of stay was on a rainy day.

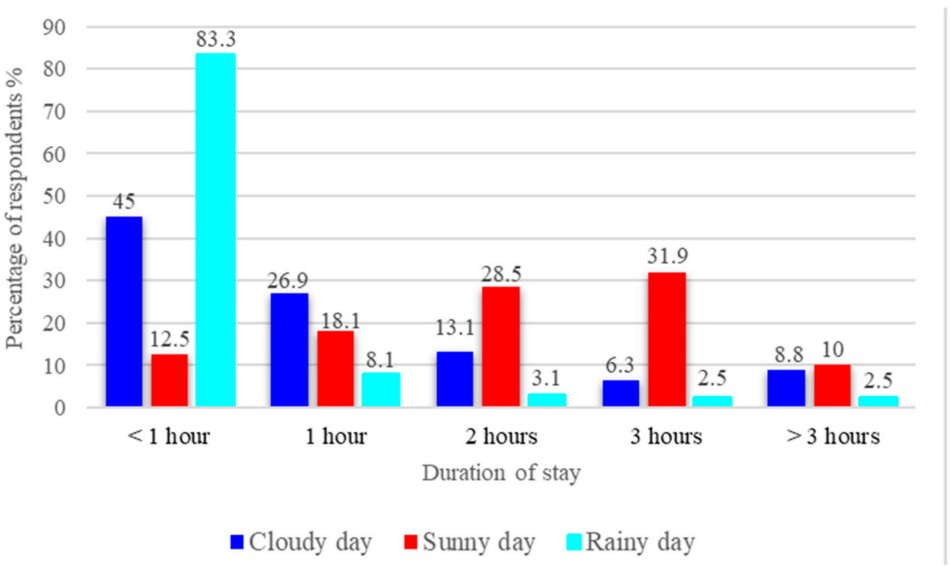

**Figure 4.** Respondent percentages of durations of stay in the park under cloudy, sunny, and rainy weather conditions.

Figure 5a–c shows the Spearman's correlation coefficient ($r_s$) of the densities of vegetation and each perceived acoustic comfort indicator. From the results, there was a strong positive correlation between low-density vegetation and pleasant acoustic comfort indicators, which were statistically significant ($r_s = 0.954$, $p < 0.01$). Calming acoustic indicators showed a strong positive correlation with low-density vegetation, which was statistically significant ($r_s = 0.925$, $p < 0.01$). There was a moderate positive correlation between high-density vegetation and the perceived acoustic indicators of annoying and chaotic, with both correlations having the same coefficients ($r_s = 0.712$, $p < 0.01$). There was a similar occurrence in medium density, where the correlation coefficients of the perceived acoustic indicators of annoying and chaotic were the same ($r_s = 0.815$, $p < 0.01$). The smallest correlation coefficient obtained was between the high-density vegetation and vibrant, which was moderately correlated and significant ($r_s = 0.617$, $p < 0.01$). In addition, there were moderate to strong positive correlations between the correlated variables.

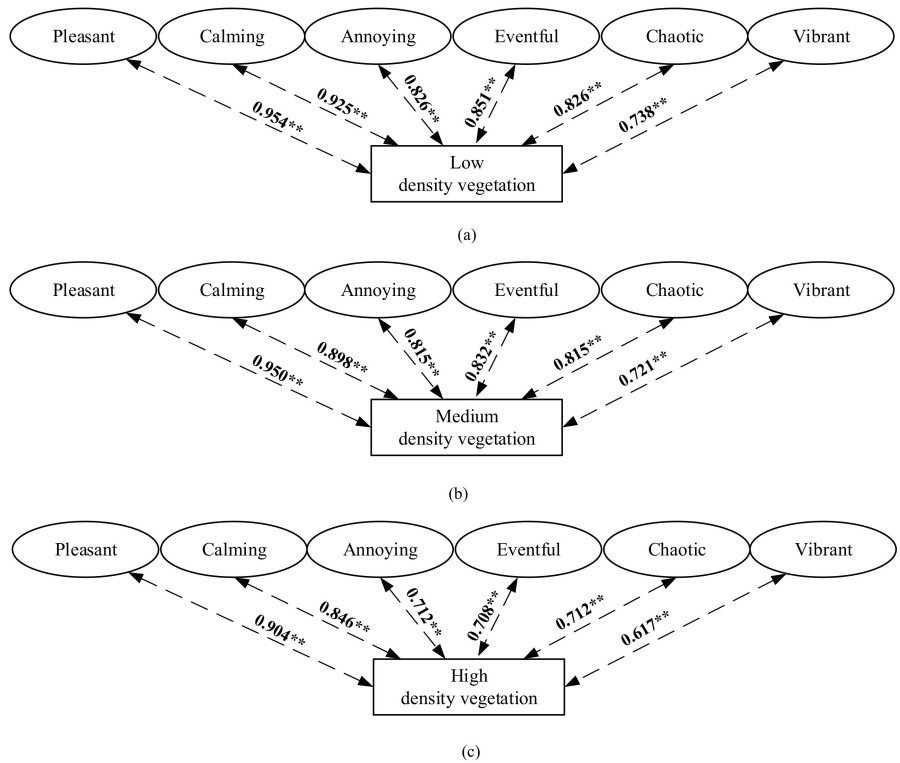

**Figure 5.** Correlations between densities of vegetation and perceived acoustic comfort indicators. (**a**) Low-density vegetation; (**b**) medium-density vegetation; and (**c**) high-density vegetation (** $p < 0.01$).

A chi-squared test was conducted between the ages of the respondents, the preference for weather conditions to visit the park, and the preference for vegetation density. The results indicated there was a strong relationship between age, preference for weather conditions ($X^2$ (15, N = 160) = 152.59, $p < 0.001$), and preference for the density of vegetation ($X^2$ (10, N = 160) = 199.69, $p < 0.001$). In addition, the results showed that there was a relationship between gender and preference for weather conditions ($X^2$ (12, N = 160) = 147.77, $p < 0.001$). The results also showed that 20.6% of the male respondents preferred cloudy days, 30% preferred sunny days, and 1.3% preferred rainy days. A total of 45.6% of female respondents preferred sunny days, and 2.5% preferred all weather conditions. Table 5 shows the influence of demographics on the respondents' preferences for the density of vegetation to be in and the weather conditions when visiting the park.

**Table 5.** Influence of the demographics of the respondents regarding preferences to weather condition and vegetation density.

| Age of Respondents | Percentage of Response (%) | Preferred Weather Condition |
|---|---|---|
| (<18, 19–30) | 20.6 | Cloudy day |
| (19–30) | 1.3 | Rainy day |
| (19–30, 31–40, 41–50, 51–60) | 75.6 | Sunny day |
| (>60) | 2.5 | All weather |

| Age of Respondents | Percentage of Response (%) | Preferred Density of Vegetation |
|---|---|---|
| (<18, 19–30) | 20.6 | Low density |
| (19–30) | 40 | Medium density |
| (31–40, 41–50, 51–60, >60) | 39.4 | High density |

## 4. Discussion

Nature and green spaces contribute to public health by reducing stress and psychological disorders [64–66]. They attract and function as a living habitat for living organisms, especially birds that produce sounds [67]. For these reasons, there has been a recent establishment of programs and agencies that are dedicated to tree planning, afforestation, and the significant control and ban of deforestation in Nigeria. As research into soundscapes continues to gain a wider spectrum in other diciplines, especially in urban planning and noise control engineering [68], it is paramount that established agencies that are responsible for urban growth continually avoid undermining the positive and health-related roles of urban forests and green spaces on the environment.

For this same reason, the National Environmental Standards and Regulations Enforcement Agency (NESREA) of Nigeria is a targeted body to which this research should be applied. The agency established in 2007 that it was committed to ensuring that Nigerians gain a feeling of being in a cleaner and healthier environment. The evaluation of avian species, as highlighted in this study, gives a better insight into the richness and density of vegetation and its effects on the sound pressure level and singing times of the birds in an environment that is meant to be conserved and protected. This study has also shown that bird calls are affected by their environment and by the weather conditions of the day. The NESREA should ensure that urban green spaces are continually conserved, as avian species could be threatened if natural vegetation in the environment becomes low. The smart and sustainable planning of a city's urban forest and green parks should be promoted using policies and guidelines which favor the economy and wellbeing of the citizens [69]. For these reasons, the Ministry of Housing and Urban Development is responsible for the creation of urban parks and recreational centers in Nigeria, and the maintenance and management of these parks is the responsibility of the Ministry of Environment. This study has shown the need for the creation of sub-agencies in various states in the country, led by the Ministry of Environment in Nigeria, to ensure that urban parks, recreational centers, and other green spaces that are responsible for providing an environment for healthy activities are constantly being evaluated. This will ensure that the acoustic environment meets international standards. In addition, the agencies should ensure that park users' perceptions of the natural environment are considered as being important and are not neglected. This study has evaluated users' perceptions of natural sounds, especially birdsongs in the park, and human preferences for weather conditions for visiting the park and the density of natural vegetation to be in. The results from this study should inform the Ministry of Environment and other agencies in Nigeria, which are responsible for providing a greener and healthier environment, to consider the provision of facilities that protect humans from harsh weather conditions in the park, especially during rain. The park's vegetation, irrespective of density, should be conserved and nurtured so that its positive environmental benefits can be reaped. Policies that allow for the provision and maintenance of a restorative environment through natural vegetation should be implemented and revised if need be, in order to meet the current needs of citizens.

This study also has its limitations, as previous studies have shown that breeding seasons affect the vocalizations of birds [70,71]. However, this study focused on vegetation densities, weather conditions, and acoustic perceptions, and thus, it did not take into consideration the breeding seasons of the birds. This will be captured in future research, where birds' behaviors during different climatic seasons of the year will be studied. In addition, this study classified five common birds in the three different vegetation densities, and the sample locations were grouped into three densities of vegetation. Future studies would entail the classification and identification of more birds in the park and would include more sample locations. The spectrogram of the birdsongs is important for determining the perceived acoustic perceptions of birdsongs [72]. Since the main purpose of the current study was to determine the effects of forest density on birdsong and human acoustic perception in an urban park, and the results were correlated with perceived acoustic com-

fort, a spectrogram was therefore not conducted, but this approach will be considered in future studies.

## 5. Conclusions

This article focused on different densities of vegetation and how these affect the sound pressure level, the frequency of singing events, and the duration of singing by individual bird species, and how this affects the human perception of sound in the natural environment.

In the context of whether different densities of natural vegetation have effects on the frequency of singing events for birds in cloudy, sunny, and rainy weather conditions, there was a significant difference between the frequency of singing events of birds on cloudy and sunny days only, with the results showing that there were no effects of vegetation densities on the frequency of singing events of birds on rainy days. With reference to the duration of singing for each species of bird, the vegetation density and the weather conditions affect the duration of singing for each bird. Some species of birds did show a different length of singing when heard in different vegetation densities under different weather conditions.

The vegetation density also affects the sound pressure levels of the birdsongs. As shown in this study, there is a significant difference between the sound pressure level of birdsongs in different densities of vegetation on a cloudy day, a sunny day, and a rainy day. In addition, the frequency of bird singing events is affected by the sound pressure levels of birdsongs, indicating that there is a significant positive correlation, as shown in Figure 3, and that the human perception of birdsongs is not affected by the sound pressure levels of the singing birds.

The respondents showed that they perceived the sound environment in the park as being pleasant, calming, eventful, and vibrant, and they did not perceive the sound environment as being chaotic or annoying. The respondents also experienced better comfort in low-density vegetation, and the least comfort in high-density vegetation. In addition, the respondents also most preferred to visit the park on sunny days, and least preferred to visit the park on rainy days. It can be seen that demographics were key factors in determining how the acoustic environment was perceived and the preferred vegetation density to be in. Regarding the length of time that people spent in the park, the respondents spent a longer time in the park on sunny days and the least time on rainy days.

The findings from this research are beneficial for the design and furnishing of public parks. It is vital to factor in different weather conditions as a design consideration when designing new parks or upgrading existing parks. In addition, the park landscape should be of high-density vegetation if birdsong is to be more frequent. In addition, parks should be furnished with facilities that would encourage people to visit and to enjoy nature, irrespective of the weather conditions, especially on rainy days where the shortest visiting time and lowest preference was recorded.

**Author Contributions:** Conceptualization, M.N. and Q.M.; data curation, M.N.; methodology, M.N., Q.M. and D.Y.; formal analysis M.N., Q.M. and D.Y.; funding acquisition, Q.M., D.Y. and F.L.; software, M.N. and Q.M.; writing—original draft preparation, M.N., writing—review and editing, M.N., Q.M., D.Y. and F.L. All authors have read and agreed to the published version of the manuscript.

**Funding:** This study was supported by the National Natural Science Foundation of China (NSFC) (grant numbers 52178070, 51878210, 51678180, and 51608147), the Open Projects Fund of the Key Laboratory of Architectural Acoustic Environment of Anhui Higher Education Institutes, Anhui Jianzhu University, Hefei, China (AAE2021ZD03), and the Natural Science Foundation of Heilongjiang Province (YQ2019E022).

**Institutional Review Board Statement:** Not applicable.

**Informed Consent Statement:** Not applicable.

**Data Availability Statement:** Not applicable.

**Conflicts of Interest:** The authors declare no conflict of interest.

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
