# Peer review of "Effects of Forest on Birdsong and Human Acoustic Perception in Urban Parks: A Case Study in Nigeria"

_forests, doi:10.3390/f13070994_

Round 1
Reviewer 1 Report
1) I recommend removing p-values in the abstract.
2) In the introduction, the authors should provide more literature reviews on the positive effects of birdsongs on soundscapes in urban areas.
3) “International Standard Organization defines it as the acoustic environment as perceived, experienced or understood by a person of people in context (ISO, 2014)” Please site the ISO proper format in the manuscript (e.g., ISO 12913-1, 2014).
4) Figure 5: Be consistent in style and format of figures (e.g., font, font size, colors, etc.).
5) In Section 4, the authors should address the potential limitations of this study.
Author Response
Reviewer 1:
I recommend removing p-values in the abstract.
Response: From line 18-23: The p-values in the abstract has been removed and abstract has been improved
In the introduction, the authors should provide more literature reviews on the positive effects of birdsongs on soundscapes in urban areas.
Response: From line 96-103: This is an amazing observation and the have gracefully authors have provided recent and more literature reviews on the positive effects of birdsongs on soundscapes in urban areas.
International Standard Organization defines it as the acoustic environment as perceived, experienced or understood by a person of people in context (ISO, 2014)” Please site the ISO proper format in the manuscript (e.g., ISO 12913-1, 2014).
Response: From line 500-502: This observation have been noted. The International Standard Organization definition of soundscapes have been sited properly.
Figure 5: Be consistent in style and format of figures (e.g., font, font size, colors, etc.).
Response: From line 351-361: The authors are thankful for the detailed observation: Figure 5 style have been corrected. The colors in figure 5 cannot be the same since they are meant to show different explanations. Cloudy day, sunny day and rainy day needed be shown in different colors in order to give the readers better clarity.
In Section 4, the authors should address the potential limitations of this study.
Response: From line 407-410: The potential limitations of the have fully been addressed.

Reviewer 2 Report
I have attached comments in my PDF.

Author Response
Reviewer 2:
All corrections pointed out by the reviewer on the manuscript have been implemented and valid explanations have been provided in the areas that were needed.
Comment: delete “Since” .line 9
Response: Line 11: This observation is kindly appreciated. The grammatical error have been effected by deleting the word “Since”
Comment: to study how? And delete how. Line 10
Response: Line 11: Thank you for that observation. Further explanation have been provided to buttress my point and grammatical errors have been addressed.
Comment: definition of acoustic comfort?. Line 14
Response: From line 17-18: This comment is important and the definition has been provided. Thank you
Comment: what is not significant? And sound pressure level? Line 17-19
Response: From line 18-23: The pointed out issue have been corrected accordingly. The significant difference between the number of singing time of birds in each density of vegetation have been stated correctly. In addition, statement of the measured sound pressure level of the birdsongs have been explained in details.
Comment: Way to many ideas in this sentence. There are three concepts strung together with very little logical connection. Line 34
Response: From line 34-39: That was a careful observation, which the authors appreciate. The sentence have been reconstructed also and recent and more literature have been provide accordingly.
Comment: What is the difference between sustainable green and sustainable. Sentence is unclear. Line 38
Response: From line: 42-43: Great observation this minor grammatical error has been corrected accordingly.
Comment: Grammatical error use order instead of other. Line 45
Response: Line: 48: Thanks for that observation it has been effected accordingly.
Comment: explain where the strategies that support architects in Nigeria is important in. Line 47-48
Response: From line 50-56: This section of the work have explained properly the importance of the established evaluation strategies which is in favour of the inhabitants and also stated the specific groups of people it caters for and how it positively promotes continuous growth and development in urban areas.
Comment: researches typically used as a verb not a noun. Line 50
Response: Line 54: Excellent observation the grammatical error have been corrected accordingly.
Comment: substitute anthropogenic. Line 59
Response: Line 62: This observation was important and has been carefully stated. As anthropogenic disturbances can be highly detrimental to human and animal population.
Comment: attracts what and functions to do achieve what? Line 67
Response from line 69-72: This observation has led to paraphrasing of that section line. It has explained with reference the importance of vegetation being part of the natural ecosystems. It further explained the positive and restorative effects exposure to vegetation provides for human beings.
Comment: “Birds” should not be capitalized in the middle of a sentence. Line 91
Response: Line 93: That observation has been noted and corrected accordingly.
Comment: Structure” Why capitalized? Line 99
Response: Line 105: That has been corrected accordingly.
Comment: in” shown in what? Line 132
Response :Line 137: That was an error that has been deleted.
Comment: This implies the trees are planted? Is this true because is contradicts the untouched natural vegetation statement. Line 138
Response: Line 141-144: The trees are not planted, that was a wrong explanation which have been deleted and corrected with the right word evergreen. Thank you.
Comment: Not a species. This is a family. Line 157
Response: Line 162: Impressive observation and such an omission has been corrected with the right word (Spizella passerina).
Comment: Migrans should not be capitalized. Line 157
Response: Line 162: this has been corrected with a lower case. (migrans)
Comment: add types. Line 159
Response: Line 164: corrected accordingly
Comment: So it seems like the goal was to figure out how loud the birds are. The question is why? How loud you are is a function of mainly how close are you to the bird, followed by how many individuals are calling but that is a very complex measurement to make and sound pressure and number of individuals are only correlated if you control for distance. Line 167
Response: from line 116-119: according to the already stated research questions on line 116-119 the goal was not to figure out just the loudness of the birds. The goal was to determine whether the density of natural vegetation has significant difference on the sound pressure level of birdsongs under different weather condition in an urban park. This would determine if when the birds are in different vegetation densities under different weather conditions (cloudy, sunny and rainy) if there is significant difference in their sound pressure levels. According to the existing body of knowledge by Traux, 2001: Pijanowski et al., 2011 factors such as time of the year, hour of the day, weather conditions and climatic dynamic are controlling factors affecting birds and birdsongs (line 103-108). How loud is function of mainly how close one is to the birds that is why as a result of the height of the trees being above 3-6 meters the height of the measuring instrument was 3 meters high to be able to have better access to the sounds. The individual birds or the number of birds was not in consideration because it would be complex to follow. And that as not the goal that is why certain species of birds that were found to be common in the tree densities of vegetation were classified.
Comment: So you mean spectral frequency (hertz) or temporal frequency (songs per unit time). Line 170
Response: Line 175: Great observation that was not frequency in hertz that was supposed to be frequency as per number of times occurred. However for better clarification to avoid further misunderstandings that word have been corrected.
Comment: Can you is a question of ability (i.e rules from park authorities on when park is open) NOT do you which is "what do you do"? This question may have created a bias. Table 1
Response Table 1: Thank you for pointing that out however, in accordance with the third research question from line 120-122 to determine the human acoustic perception in the park. We used the word can because in English language according to Britannica dictionary it is a polite way to ask a question. Since visiting the park, being in the park for a certain period of time and visiting the park in different weather conditions was a matter of choice by the citizens of Abuja the capital city of Nigeria. We felt the use of can was appropriate to ask the questions. Thank you.
Comment: So I think this analysis is very biased. If your goal was to determine "on average" how loud are birds at a particularly time of day and place you would need a random sample from multiple locations NOT a sample as you described where you find the birds and measure how loud they are. How many places in each vegetation density were there, how often did they occur etc. This is important because an ANOVA is the wrong test. You have a repeated measures ANOVA at a minimum. Line 194
Response: Thank you for your opinion. With that, we would gladly throw more light on our research goal. The goal was not to determine particularly on the average how loud the birds are. It was whether the density of natural vegetation has significant difference on the sound pressure level of birdsongs under different weather condition in an urban park. This brings us to understand if the different vegetation in which they stay in affects their loudness and this was considered in different weather conditions as earlier stated. The sample was from multiple locations in the selected vegetation densities. This was explained in methods of recording and method of measurement stating that the instruments were place in different locations of the park explained in line 165-171. That being said this brings us to clarify that the analysis was not bias. Also in accordance with the research aim, the Analysis of Variance (ANOVA) was appropriate because it would tell if there are any statistical differences between the means of three or more independent group. Since the average of the measured sound pressure level of birdsongs was measured in the three vegetation density in different weather conditions, this leads us to having three groups of data for analysis and in line with the aim it was not biased to use one way analysis of variance.
Comment: Is this the number of times the birds sang? Very unclear. Line 200
Response: Yes it clearly is.
Comment: Is this a separate analysis for each weather condition. This is invalid analysis technique as stated above and breaking it up seems illogical and would be a two-factor repeated ANOVA as described. Line 210
Response: Line 211-216: This is an analysis to will not be regarded as invalid because of the aim of the research. It was aimed at considering weather conditions as a factor. Therefore, analysis technique that puts different weather conditions into consideration was needed.
Comment: This is the first time it became clear that you measured singing/ calling rate (songs per unit time). Be explicit in your description of response variables. Line 249
Response: Thank you for the observation it was the further information we provided to add more value to the paper.
Comment: Of course this is true. It is not required as an analysis. When there is sound, sound pressure level will be higher. Line 289
Response: Line 288-293: We understand the point stated; however, the essence was to provide further analysis to know and clearly buttress the point that there is a strong relationship between the number of singing time of birds in different density of vegetation and the sound pressure when they are under different weather conditions. “When there is sound, there will be sound pressure level yes”: But this analysis is stating that the number of times birds calls will influence the loudness of the call, which is affected by factors like vegetation density it finds itself and the weather conditions amongst other already stated factors.
Comment: The metric of sound pressure level and duration of time it was collected over are fundamental to making any sense of this as an analysis. Line 292
Response: Line 165-167: The sound pressure level was measured for a period of 15 minutes interval within one hour for a total period of 8:00-18:00. For example from 8:00-8:15-8:30-8:45:9:00 etc.
Comment: Measuring sound pressure level when a bird is singing at second level will create a perfect correlation, at a minute level then other sources of noise could be part of SPL, at longer time scales this correlation will be weaker. My point being how you sampled and what your goal of this metric is need to be stated far more clearly. Line 292
Response: Line 166-171: Thanks this has been described and explained. Random sampling and placing the instruments at height 3.0 meter to get closer to the birdsongs. This was done to take care of reduce the interference of other noise sources as this will mean being closer to the sound source.
Comment: You stated this already with sample size. Line 324
Response: line 325-326: This was stated to give more clarity and more information to the reader.
